# Differences in the Tumor Molecular and Microenvironmental Landscape between Early (Non-Metastatic) and De Novo Metastatic Primary Luminal Breast Tumors

**DOI:** 10.3390/cancers15174341

**Published:** 2023-08-30

**Authors:** Yentl Lambrechts, Sigrid Hatse, François Richard, Bram Boeckx, Giuseppe Floris, Christine Desmedt, Ann Smeets, Patrick Neven, Diether Lambrechts, Hans Wildiers

**Affiliations:** 1Laboratory of Experimental Oncology (LEO), Department of Oncology, KU Leuven, 3000 Leuven, Belgium; 2Laboratory for Translational Breast Cancer Research (LTBCR), Department of Oncology, KU Leuven, 3000 Leuven, Belgium; 3Laboratory of Translational Genetics, Department of Human Genetics, VIB-KU Leuven, 3000 Leuven, Belgium; 4VIB Center for Cancer Biology, 3000 Leuven, Belgium; 5Laboratory for Cell and Tissue Translational Research, Department of Imaging and Radiology, KU Leuven, 3000 Leuven, Belgium; 6Department of Pathology, University Hospitals Leuven, 3000 Leuven, Belgium; 7Department of General Medical Oncology, Multidisciplinary Breast Center, University Hospitals Leuven, 3000 Leuven, Belgium; 8Department of Surgical Oncology, University Hospitals Leuven, KU Leuven, 3000 Leuven, Belgium

**Keywords:** (luminal) breast cancer, breast cancer biology, mutations, primary tumor, de novo–stage IV, tumor microenvironment

## Abstract

**Simple Summary:**

In de novo metastatic luminal breast cancer, there remains an urgent need to fill the knowledge gap about which molecular mechanisms drive de novo metastatic disease. This study is the first to explore the transcriptomic profile and tumor microenvironmental differences at baseline of patients with ER+/HER2− de novo metastatic luminal breast cancer compared to patients with early (non-metastatic) luminal breast cancer to determine which aspects of the tumor microenvironment of de novo luminal breast cancer are altered and obtain perspective of the molecular mechanisms underlying breast cancer metastasis.

**Abstract:**

**Background:** The molecular mechanisms underlying the de novo metastasis of luminal breast cancer (dnMBC) remain largely unknown. **Materials and Methods:** Newly diagnosed dnMBC patients (grade 2/3, ER+, PR+/−, HER2−), with available core needle biopsy (CNB), collected from the primary tumor, were selected from our clinical–pathological database. Tumors from dnMBC patients were 1:1 pairwise matched (n = 32) to tumors from newly diagnosed patients who had no distant metastases at baseline (eBC group). RNA was extracted from 5 × 10 µm sections of FFPE CNBs. RNA sequencing was performed using the Illumina platform. Differentially expressed genes (DEG)s were assessed using EdgeR; deconvolution was performed using CIBERSORTx to assess immune cell fractions. A paired Wilcoxon test was used to compare dnMBC and eBC groups and corrected for the false discovery rate. **Results:** Many regulatory DEGs were significantly downregulated in dnMBC compared to eBC. Also, immune-related and hypoxia-related signatures were significantly upregulated. Paired Wilcoxon analysis showed that the *CCL17* and neutrophils fraction were significantly upregulated, whereas the memory B-cell fraction was significantly downregulated in the dnMBC group. **Conclusions:** Primary luminal tumors of dnMBC patients display significant transcriptomic and immunological differences compared to comparable tumors from eBC patients.

## 1. Introduction

Improved screening programs and treatment strategies have strongly decreased breast cancer mortality rates. However, hormone-sensitive luminal breast tumors, which are estrogen receptor (ER) positive, progesterone receptor (PR) positive or negative, and human epidermal growth factor receptor 2 (HER2) negative, still represent a challenging subtype for oncologists, especially the more aggressive, highly proliferative so-called luminal B-like subtype, which is associated with a poorer prognosis than the more quiescent luminal A-like [1,2]. While luminal A-like breast cancer can often be adequately treated with surgical resection of the tumor and subsequent anti-hormone therapy, treatment of the luminal B-like tumor type may demand a more rigorous treatment regimen with (neo-)adjuvant systemic chemotherapy to decrease the risk of future relapse and development of distant metastasis [3,4,5]. 

Moreover, a subpopulation (±5%) of patients with luminal B-like breast cancer presents with de novo metastatic or stage IV disease at initial diagnosis. This patient population is considered a poor prognostic group with incurable disease. De novo metastatic breast cancer (dnMBC) is managed in a different way than early (non-metastatic) breast cancer (eBC): unless there are very few and resectable metastatic lesions (also referred to as oligometastatic disease), the primary tumor is not surgically removed, and patients only receive systemic treatment. Although recurrent and de novo metastatic patients are administered comparable systemic (chemo)therapies, they differ in molecular patterns [6]. Seltzer et al. reported that dnMBC has an increased frequency of *PTEN*, *ABL2*, and *GATA3* mutations, together with downregulated *TNFα*, *IL-17* signaling, and chemotaxis, as compared to recurrent metastatic breast cancer. In addition, they found an upregulation in the dnMBC of steroid biosynthesis, cell migration, and cell adhesion [7]. 

A considerable number of individual genes (e.g., *TP53*, *CDKN2A*, *PTEN*, *PIK3CA*, and *RB1*), microRNAs (e.g., miR-10b, miR-21, miR-200 family, and miR-29), and chemokine ligand/receptor pairs (e.g., CXCL12/CXCR4) have been linked to the metastatic process, yet the global picture remains obscure [8,9,10]. In particular, it is unclear which molecular mechanisms drive de novo metastatic disease: why are some tumors already metastasized at diagnosis, while other breast tumors with similar biological characteristics in terms of size, grade, histology, receptor status, and lymph node involvement only spread at a later stage (i.e., after initial treatment) or do not spread at all? Also, some patients may present with small tumors that readily form metastases, while others may harbor much larger tumors of comparable grade and histology yet without metastases. These clinical observations demonstrate that tumor metastasis is not just a matter of time and size. Although metastasis has long been considered as a late event in cancer progression, it has become clear over the past decennia that tumor cells can disseminate from very premature, even pre-neoplastic lesions [11]. Rhim et al. showed in a mouse model of pancreatic cancer that pre-neoplastic tumor cells can undergo epithelial–mesenchymal transition (EMT) and disseminate into distant organs before a primary tumor could even be detected [12]. Another study using an inducible expression of oncogenes in mammary epithelial cells revealed that untransformed mammary epithelial cells can survive as disseminated cells in the lung and start malignant proliferation upon oncogene activation [13]. Dormant tumor cells that have already seeded a secondary organ, but have not yet grown out to overt metastases, are frequently found in cancer patients [14]. As a result of the diverging temporal dynamics between initial tumor cell dissemination and the later outgrowth of metastatic lesions, many cancer patients may be diagnosed with early-stage cancer and undergo systemic treatment with curative intent yet subsequently succumb to distant metastatic relapse. It is also important to note that due to the limitations of standard staging procedures, de novo metastasis probably remains undetected in a significant proportion of cases at breast cancer diagnosis. 

Very few studies have investigated the biological/molecular differences between primary tumors from patients with dnMBC compared to breast tumors from patients with eBC. We have set up such a study in order to disclose the tumor molecular pathways involved in the metastatic cascade and to explore potential distinct treatment options for these two patient populations. To this end, we compared the tumor transcriptomic profiles of breast tumors from patients with ER+/HER2− (which is the most frequent breast cancer subtype) dnMBC and pair-wise matched breast tumors from patients with eBC. 

## 2. Material and Methods

### 2.1. Patient Population

The Leuven Multidisciplinary Breast Center (Leuven, Belgium) at the University Hospitals Leuven’s institutional clinicopathological database was used to select the appropriate patients. In-depth patient and tumor characteristics and follow-up data are all documented in this database.

Selected patients met the following criteria: (i) newly diagnosed between 2004 and 2019 with dnMBC breast cancer, grade II or III invasive breast carcinoma of non-special type (IBC-NST) (other breast cancer subtypes, grade I IBC-NST, multifocal, and bilateral tumors were not allowed to avoid important population heterogeneity); (ii) tumor being ER positive and HER2 negative (ER positivity was defined as at least 1% of cells staining positive according to ASCO-CAP guidelines [15], HER-2 positivity was defined according to ASCO-CAP 2018 guidelines [16]); (iii) cTNM staging available. The staging was completed according to local standards: in general, for clinical stages I–II, a chest Rx, liver ultrasound and bone scintigraphy were performed, while for clinical stages III–IV, a CT or PET-CT were performed as per standard of care. CA15.3 is also routinely assessed in all new invasive breast cancer patients. In case CA15.3 was significantly increased, or suspicious lesions (potential metastases) were found on clinical examination and/or imaging, further investigations were performed to confirm de novo metastatic disease; (iv) core needle biopsy at the time of primary diagnosis available, with sufficient remaining tumor tissue (surgical resection specimens were not allowed for non-metastatic patients, to avoid technical bias); (v) no prior invasive breast cancer.

After selecting the patients with de novo metastatic disease (dnMBC), a 1:1 matching eBC group was created, consisting of patients carrying breast tumors with similar characteristics but without de novo metastases at diagnosis. Patient matching was based on age at diagnosis as the first criterium, since age is an important confounder: older patients present with larger tumors, which exhibit different biological properties [17], and lymph node involvement behavior [18]. Tumor grade was applied as the second criterium (grade 2 or grade 3) given the important prognostic value of tumor grading. Thirdly, patients were matched for clinical tumor staging (cT1, cT2, cT3, cT4, cT4b, cT4c, and cT4d) and lymph node involvement (cN0, cN1, cN2, and cN3) 

### 2.2. Pathologic Assessment of Hematoxylin and Eosin (H&E)-Stained Tumor Slides and RNA Extraction

RNA extraction was performed on five consecutive sections of formalin-fixed paraffin-embedded (FFPE) core needle biopsies cut at 10 µm thickness and flanked by a hematoxylin and eosin (H&E)-stained slide to control the representativity of the sample and the relative percentage of tumor cells, stromal components, and inflammatory infiltrate by an expert breast pathologist (G.F.). Shortly after, the relative proportion of malignant epithelial cells, normal epithelial cells, mononuclear inflammatory cells, and fibroblasts was completed by eyeballing on 10×–20× magnification, scanning the whole slide, and expressing as a percentage of the total cellular (benign + malignant) population present in the sample. Additionally, the mononuclear inflammatory cells infiltrating the stroma adjacent to the tumor cells were scored using the standardized scoring method proposed by the tumor-infiltrating lymphocyte (TIL)s international working group [19]. Plasma cells identified on H&E were scored using a semiquantitative grading system previously published [20]. The HighPure FFPET RNA extraction kit (Roche^®^, Basel, Switzerland) was used following the manufacturer’s protocol. After extraction, RNA concentrations were measured on the NanoDrop^®^ 2000 UV Visible Spectrophotometer (Thermo Scientific™, Waltham, MA, USA), and quality (RNA integrity) was assessed on the Aligent Bioanalyzer. 

### 2.3. RNA Sequencing

RNA libraries were created using the Illumina TruSeq RNA sample preparation kit V2 according to the manufacturer’s instructions, and resulting whole-exome libraries were sequenced on a HiSeq2500 or HiSeq4000 (Illumina), generating 50 bp reads. After the removal of adaptors and optical duplicates, the raw sequencing reads were mapped to the human transcriptome GRCh37 using TopHat 2.0 and Bowtie 2.0 [21]. Reads were assigned to ensemble gene IDs with the HTSeq software package (version 0.6.1p1). On average, 28.7 × 10^6^ ± 10.9 × 10^6^ reads were assigned to genes. These reads were normalized with EDASeq [21]. 

### 2.4. Bioinformatic Analysis

Differential expression was assessed with EdgeR [22], and gene-set enrichment scores for the hallmark pathways were calculated with gene set variation analysis (GSVA) [23]. Gene signatures were retrieved from the literature: Gene21 [24] (proxy to Oncotype Dx), Gene70 [25] (proxy to MammaPrint), genomic grade index (GGI) grading [26], continuous hypoxia [27], cyclic hypoxia [27], IFNA [28], IFNG [28], and AKT-MTOR-MG [29], and they were computed as a weighted average of the normalized gene expression with coefficients equal to 1 or −1 for genes positively and negatively associated with the signature, respectively.

Tests for differentially expressed genes (DEG)s were corrected for multiple testing using the Benjamini–Hochberg method. Immune cell fractions were evaluated using the CIBERSORTx software (available from https://cibersortx.stanford.edu/) with 1000 permutations in absolute mode with batch correction on normalized expression data using the signature matrix “LM22” [30]. A paired Wilcoxon test was used to compare immune-related gene signatures and cell fractions between dnMBC and eBC.

Gene ontology (GO) analysis using Gorilla [31] was performed on DEGs between eBC and dnMBC groups (*p*-value threshold of 10^−3^). GO analysis was performed in the *Homo Sapiens* organism using two lists of genes (target list—DEGs; background list—all genes present within our cohort) for the three compartments (biological process, molecular function, and cellular components). Thereafter, all the significant GO terms for all three compartments were semantically summarized and visualized in the web tool REVIGO [32] using default parameters except for the species parameter set to *Homo Sapiens*.

### 2.5. Statistical Analysis

Associations between clinicopathological variables and group (dnMBC, eBC) were assessed with Fisher exact tests. Associations between continuous scores (i.e., pathological scores, gene signatures) and group were assessed either with paired Wilcoxon tests or regressions. For the latter, linear mixed models were built with the continuous scores as outcome, the group (dnMBC vs. eBC) as a fixed effect, and a random effect on the patient pair. Covariates were also added in the multivariable models such as cT (cT4, cT3, cT2 vs. cT1), cN (positive vs. negative), and grade (3 vs. 2). Given the apparent association between the group (eBC vs. dnMBC) and cN, we emphasized model 4, which is the model corrected for cT and grade, reflecting the disease’s aggressiveness. Estimates, 95% confidence intervals, and Wald test *p*-values were computed using the nlme R package (version 3.1.161). When required, *p*-values were adjusted for multiple testing using the Benjamini–Hochberg method. Analyses were performed with R 4.2.1.

## 3. Results 

### 3.1. Patient and Tumor Characteristics 

The 1:1 matched cohort consisted of 32 patients with de novo metastasized breast cancer (dnMBC group) and 32 patients with non-metastasized early breast cancer (eBC group) with ER+ HER2- tumors. Overall, 87% (n = 23) was PR+ in the dnMBC group and 94% (n = 30) was PR+ in the eBC group. Median ages were 62 years and 61 years for dnMBC and eBC, respectively. Most of the patients presented at diagnosis with a grade 3 tumor (dnMBC: 56% vs. eBC: 53%). The median clinical tumor size was 37 mm in the dnMBC group versus 27 mm in the eBC group, and lymph node involvement was also more present in the dnMBC group. Matching for age and grade was adequate but suboptimal for cT and cN. This was due to the fact that the pool of eligible patients for selecting the matched eBC cases was further narrowed by tissue availability: we could only work with patients with sufficient residual tumor tissue from diagnostic core biopsies. Primary metastatic disease, in this study the dnMBC group, is nearly never operated, so we needed to use core biopsies from both groups (omitting resection specimens from eBC patients) to avoid technical bias. When looking at the location of metastasis within the de novo metastasized group, metastasis to the bone appears to be predominant (66% of cases). All the patients’ characteristics are summarized in Table 1 and Figure 1A. 

### 3.2. De Novo Metastasized (dnMBC) and Non-Metastasized Breast Tumors (eBC) Exhibit Comparable Cellular Composition

Tumor cellular characteristics are indicated in Figure 1B and Appendix A. No significant differences were noted between the dnMBC group and the eBC group with regard to the presence of TILs (median: 2.0% vs. 2.0%; *p* = 0.760), mononuclear inflammatory cells (median: 10.0% vs. 10.0%; *p* = 0.480), plasma cells (based on scoring, *p* = 0.130), tumor epithelial cells (median: 50.0% vs. 50.0%; *p* = 0.300), normal epithelial cells (median: 2.5% vs. 0.0%; *p* = 0.220), and fibroblasts (median: 30.0% vs. 35.0%; *p* = 0.100), which is in agreement with the multivariable analysis corrected for cT and grade (Appendix A and Appendix A). These results confirm that both groups’ core needle biopsies are similar in terms of cellularity and can be used for comparative transcriptomic analysis.

### 3.3. Gene Expression Signatures Did Not Differ between De Novo Versus Non-Metastasized Tumors

We have performed a paired Wilcoxon comparison of both groups for several commonly used gene expression signatures, such as GENE21, GENE70, and a gene expression signature from the GGI. There was no significant difference between the dnMBC and the eBC tumors with regard to these gene expression signatures GENE21 (*p* = 0.410), GENE70 (*p* = 0.095), and GGI (*p* = 0.720) (Figure 1C), which is in agreement with the multivariable analysis corrected for cT and grade (Appendix A and Appendix A). This indicates that tumors from both groups cannot be distinguished based on the classical (e.g., proliferation-related) risk factors integrated with these prognostic tests.

### 3.4. Tumor Microenvironment Differs at the Time of Diagnosis

First, we looked into the GSVA hallmark signatures [23]. No hallmark signature was found statistically significant after FDR correction (Appendix A). Thereafter, we explored cell fraction changes using CIBERSORTx [30]. Lastly, we investigated differences at the individual gene level. 

#### 3.4.1. Hypoxia Pathways Are Upregulated in De Novo Metastasized Tumors

Figure 2B,C represents the integrated boxplots representing the signatures “Cyclic. Hypoxia.up” and “Continuous. Hypoxia.up” that were significantly more expressed in the dnMBC group, with *p* = 0.015 and *p* = 0.045, respectively. DEGs are depicted in the volcano plot (Figure 2A) and listed in Appendix A. Here, significant upregulation of the hypoxia-inducible factor-1 alpha (*HIF-1α*) in the de novo metastasized tumors is demonstrated (*p* < 0.001). In addition, matrix metalloproteinase 2 (*MMP2*) (*p* < 0.001) was upregulated in de novo metastasized compared to non-metastasized tumors. In addition, we noted an upregulation of *P4HA1* (*p* = 0.013) and *PLOD2* (*p* = 0.020) hydrolases, and a lysyl oxidase family member, such as *LOX* (*p* = 0.005). Furthermore, the hypoxia-controlled gene *VEGF-C* was also increased in the de novo metastasized tumors (*p* = 0.021). Interestingly, the expression of *ZEB1* was also significantly elevated in de novo metastasized tumors (*p* = 0.0197). After correction for cT and grade by multivariable analysis, the global tendency toward increased hypoxia in dnMBC (as indicated by the positive coefficients) could still be noticed (Appendix A and Appendix A). Next, we performed a GO enrichment analysis from our DEGs with the online Gorilla tool, extracted the GO terms, and inserted it in the REVIGO visualization tool. Here, we could confirm that the mechanisms involved in hypoxia, such as the regulation of angiogenesis, vasculature development, extracellular matrix (ECM) disassembly, endothelial cell proliferation, cell proliferation in bone marrow, blood vessel cell proliferation involved in sprouting angiogenesis, and blood vessel endothelial cell migration, were involved in the de novo metastasized group (Figure 2D). 

#### 3.4.2. De Novo Metastasis Is Associated with an Altered Immune Landscape

To investigate changes in the immune microenvironment of de novo metastasized tumors, in comparison with their non-metastasized counterparts, we first examined the signatures IFNA.down and IFNG.down. These comprise the genes that contribute negatively to the global IFNα and IFNɣ signatures, respectively. We observed an increased downregulation of both pathways in dnMBC as compared with eBC, implying a lower anti-tumor immune response in dnMBC (Figure 3C–F). A paired Wilcoxon test showed a significant upregulation of both interferon ‘down’ signatures and of *CCL17* in de novo metastasized tumors compared to non-metastasized tumors (*p* = 0.016, *p* = 0.002, and *p* = 0.050, respectively) (Figure 3B). In addition, when performing paired statistical analysis in the CIBERSORTx software, the memory B-cell fraction was significantly lower (*p* = 0.001) whereas the neutrophil cell fraction was significantly higher in the de novo metastasized compared to the non-metastasized tumors (*p* = 0.030; Figure 3D,E). Other immune cell populations analyzed in the CIBERSORTx software were not significant (Appendix A). Further elaborating on the higher neutrophil fraction in de novo metastasized tumor tissue, we investigated if the neutrophil-to-lymphocyte ratio (NLR) in blood at baseline was significantly different in dnMBC vs. eBC (Appendix A). Baseline NLR was indeed higher in the dnMBC group (median = 3.55, average = 3.46) compared to the eBC group (median = 2.55, average = 3.16), but this trend did not reach statistical significance (*p* = 0.170). In addition, we further looked into the individual DEGs (Figure 3A; Appendix A) and noticed that many chemokines were upregulated in the de novo metastasized group, such as *CXCL14* (*p* = 0.002), *CXCL13* (*p* = 0.003), *CXCL11* (*p* = 0.007), *CXCL10* (*p* = 0.049), *CXCL9* (*p* < 0.001), *CXCL8* (*p* = 0.003), *CXCL2* (*p* = 0.016), *CCL19* (*p* = 0.023), and *CCL11* (*p* = 0.023). On the other hand, several chemokine receptors were significantly downregulated in the de novo metastasized group, including *CCR5* (*p* = 0.018), *CCR6* (*p* = 0.030), and *CCR10* (*p* = 0.003). Given that many cyto/chemokines are interrelated with the downstream effects of the MAPK pathway, our results also show that the signature AKT-mTOR is upregulated in the de novo group (*p* = 0.009; Figure 3H). Furthermore, interleukin-related factors, such as *IL-6* (*p* = 0.003), *IL6ST* (*p* < 0.001), *ILF2* (*p* < 0.001), *IL1R1* (*p* < 0.001), *IL1R2* (*p* = 0.025) and *IL13RA1* (*p* < 0.001) showed increased expression in de novo metastasized tumors. Interleukin signaling is closely connected to the JAK/STAT signaling pathway, of which several components were upregulated as well in the de novo metastasized group in our analysis, including *STAT1* (*p* = 0.029), *STAT3* (*p* = 0.001), and *JAK1* (*p* < 0.001). In agreement with the above-mentioned CIBERSORTx findings concerning memory B cell representation in the transcriptomic profile, we found that cluster of differentiation (CD) genes that are associated with memory B cells were significantly downregulated in the de novo metastasized group, more specifically *CD19* (*p* = 0.017), *CD80* (*p* = 0.022), *CD27* (*p* = 0.025), and *CD40* (*p* = 0.004). After correction for cT and grade by multivariable analysis, similar increasing or decreasing trends were still apparent for the majority of the CIBERSORTx immune cell types, paired signatures, and DEGs (Appendix A and Appendix A). Finally, the GO enrichment analysis, visualized in REVIGO, confirmed that there is an immunity-related role in the de novo metastasized group, as reflected by the GO terms: humoral immune response, immunoglobulin production, regulation of cytokine production involved in inflammatory response, regulation of interleukin-1 production, regulation of interleukin-1 beta production, and regulation of interleukin-6-mediated signaling pathway (Figure 3G).

#### 3.4.3. Numerous Regulatory Genes Are Affected in the De Novo Metastasized Tumors

Among the DEGs, as identified by their Benjamini–Hochberg value below 0.05, there were many regulatory genes, such as 14 small nucleolar RNAs (snoRNAs), 21 microRNAs, and 23 pseudogenes, which were all highly significantly downregulated in the de novo metastasized group (Figure 4A–C and Appendix A). After correction for cT and grade by multivariable analysis, similar trends (in terms of up- or downregulation) were noticed for most of the regulatory DEGs, with many of them still showing high statistical significance (Appendix A). Many aspects of the RNA regulatory compartment also emerged from the GO enrichment analysis, such as miRNA-mediated gene silencing, RNA-mediated gene silencing, post-transcriptional gene silencing, post-transcriptional regulation of gene expression, regulation of translation, ncRNA-mediated, and RNA processing (Figure 4D). 

## 4. Discussion

This study has investigated the difference in transcriptomic profiles of newly diagnosed de novo metastasized (dnMBC) versus non-metastasized (eBC) luminal breast tumors. Whereas no marked changes were disclosed in the classical prognostic gene expression signatures like GENE21, GENE70, and GGI, nor in the cellular composition of the tumor specimens used, the transcriptomic analysis clearly pointed to several aspects of the tumor microenvironment that are significantly altered when comparing de novo metastasized and non-metastasized tumors, even after multivariable analysis with correction for cT and grade (Figure 5). 

More specifically, we found that hypoxia is more prominent in de novo metastasized tumors compared to their non-metastasized counterparts. Hypoxia within the tumor occurs as a result of massive tumor cell proliferation and associated oxygen demand. On the other hand, oxygen availability is decreased due to the abnormal structural and functional vasculature that forms within solid tumors [33]. Cancer cells respond to this oxygen shortage by overexpressing hypoxia-inducible factors, such as HIF-1α, which regulates a large number of target genes involved in invasion, extravasation, and EMT [34]. For instance, Gilkes and Semenza described that invasion occurs through the degradation of the ECM component by HIF-1α-dependent MMPs, like MMP2 and MMP9 [34]. These are endopeptidases that degrade type IV collagen, and increased levels of intra-tumoral MMP2 were shown to be associated with poor prognosis. In addition, they described that HIF-1α plays a critical role in collagen biogenesis in breast tumors by upregulating the expression of P4HA1, P4HA2, PLOD1, and PLOD2 hydroxylases as well as the lysyl oxidase family members LOX, LOXL2, and LOXL4. HIF-1α activation modulates ECM synthesis to create a rigid microenvironment that improves cell adhesion, elongation, and motility [34,35]. In agreement with this, our study showed a significant upregulation of *HIF-1α*, together with several of its target genes, including *P4HA1*, *PLOD2*, and *LOX*, in primary metastasized tumors. In addition to ECM degradation, angiogenesis is also crucial for the growth and metastasis of solid tumors, such as breast tumors [34]. HIF-1α plays a vital role in the expression of VEGF under hypoxic conditions, which is also reflected in our results. Another interesting finding from our transcriptomic study is that *ZEB1* is upregulated in de novo metastatic tumors. ZEB1 is a transcriptional repressor with a potential role in initiating bone metastasis [33]. This seems consistent with the predominance of bone metastases in dnMBC patients in our study.

Secondly, we found that many immune-related genes and pathways were significantly altered in the dnMBC group. Chemokines and their receptors have versatile functions: besides sustaining the growth and survival of tumor cells, they are essential in the metastatic process to direct and promote the migration of leukocytes as well as cancer cells through the MAPK/ERK signaling pathway. We found that numerous chemokine genes were upregulated, whereas several chemokine receptors were downregulated in de novo metastasized tumors. When expressed by tumor cells, chemokine receptors can guide tumor cells to particular anatomic sites to form metastases through interaction with their cognate chemokine ligands produced at distant locations. Circulating tumor cells are thus attracted into a “premetastatic niche”, which provides a favorable setting for the development of a metastatic lesion [36,37]. This mechanism has been proposed as a potential explanation for the organ-specific metastasis patterns of distinct cancer types. It would be interesting to assess the expression of chemokines receptor genes in metastatic lesions as well, but this was not within the scope of the present study. Chemokines also recruit different immune cell subsets into the tumor microenvironment, thus mediating the tumor immune response which may influence cancer progression [37,38]. In our study, the most significant upregulated DEG within the immune compartment in dnMBC is chemokine *CXCL13*. A breast cancer study reported that the overexpression of CXCL13 in both sera and breast tumor tissues implied that CXCL13 might play a role in breast cancer initiation and progression [39,40]. Furthermore, CXCL13 is known as a B-cell-attracting chemokine (BCA-1). It was shown that a high representation of B cells in the tumor microenvironment is related to better survival in breast cancer patients [38]. Surprisingly, however, we found that memory B-cells were significantly downregulated in dnMBC tumors. Memory B-cells drive the immune response because they have B-cell receptors with a high affinity that react quickly to antigen reactivation. By acting as antigen-presenting cells, they can also contribute to activating T cells [41]. Among the chemokine family, *CXCL8* also showed significantly increased expression in de novo metastasized breast tumors. This chemokine indirectly stimulates angiogenesis by targeting and supporting the survival of vascular endothelial cells via modulation of the PI3K/MAPK pathway [38]. This signaling cascade in turn promotes downstream genes like *AKT* and *mTOR*. Accordingly, *CXCL8*, *AKT* and *mTOR* were all significantly increased in dnMBC compared to eBC in our study [42]. In addition, CXCL8 can recruit neutrophils that affect several metastasis-specific processes in the tumor, such as migration, invasion, and angiogenesis, which is in line with our observation of increased neutrophils in the dnMBC group [38]. Several studies have used NLR to assess the inflammatory status of a patient [43,44]. This NLR is considered a prognostic factor in cardiovascular diseases and multiple types of cancer [43,44]. In breast cancer, a meta-analysis by Wei et al. suggested that NLR is a good prognostic marker, with patients with high NLR having a poor prognosis [45]. In our study, NLR was not significantly different between dnMBC versus eBC patients. Furthermore, we noted that *CCL17* was significantly upregulated in de novo metastatic tumors compared to non-metastatic tumors. CCL17 is also known as ‘thymus and activation-regulated chemokine’ (TARC). A murine study of hepatocellular carcinoma indicated that T-regulatory cells are attracted by CCL17 through the CCR4 axis and that high CCR4 expression is positively associated with metastasis [46]. In addition, CCL17 seems to connect with neutrophils, as demonstrated by Mishalian and colleagues. They found that the level of CCL17 is associated with an increased abundance of tumor-associated neutrophils [47]. The increased expression of *CCL17* concurrent with an increased abundance of neutrophils in the de novo metastatic group of our study also supports this hypothesis. All these findings suggest that chemokines may be tightly interconnected with the altered presence of specific immune cell subtypes in the tumor microenvironment in dnMBC tumors. Furthermore, the increased expression of IFNA.down and IFNG.down signatures observed in our study in dnMBC may point to decreased interferon signaling, which may have a direct impact on the local anti-tumor immune response. Lastly, the JAK/STAT pathway seems to be involved as well in de novo metastasis, as indicated by an upregulation of *STAT1*, *STAT3*, and *JAK1* in dnMBC. In addition, the interrelated interleukin signaling *IL-6* and interleukin-related factors (*IL6ST*, *ILF2*, *IL1R1*, *IL1R2*, and *IL13RA1*) were also significantly upregulated in dnMBC. Interestingly, the JAK/STAT pathway is reported to be essential for the progression/development of breast cancer bone metastases [48], which is consistent with the fact that the majority of the dnMBC patients in our cohort had bone metastasis at the moment of diagnosis.

In addition to increased hypoxia and altered immune pathways, we also found many regulatory genes to be significantly downregulated in the de novo metastasized group. Of note, the expression of microRNAs in tumor cells has been shown to be decreased by cytokines produced in the inflammatory environment of cancer. For example, in colorectal cancer cells, miR-34a is downregulated by the pro-inflammatory IL-6 [49]. In addition, transforming growth factor (TGF)-β, an immune-suppressing cytokine in the microenvironment of breast cancer, inhibits members of the miR-200 family, which inhibit tumor invasion and metastatic dissemination by targeting the EMT inducing transcription factor ZEB-1, which is involved in the hypoxia-regulated microenvironment and bone metastasis [50,51]. MiR-200 and miR-34a are downregulated in our dnMBC cohort, which could predict that the inhibition of the EMT is lost, and thus, EMT is more pronounced in dnMBC tumors. In the article of Liu et al., TGF-β is also linked to miR-425, a crucial suppressor of EMT, and the development of TNBC through the inhibition of the TGF-β/SMAD3 signaling pathway [52]. Another EMT-suppressing microRNA is miR-29, which targets a network of pro-metastatic genes, such as *LOX*, *MMP2*, and *VEGF* [9,53]. MicroRNAs can also be involved in the MAPK/PI3K/AKT signaling pathways, as described in particular for the let-7/miR-98 family and miR-10a [51,54]. In our findings, these tumor-suppressing microRNAs (miR-425, miR-29, miR-98, and miR-10a) are all significantly downregulated in dnMBC tumors. In addition to microRNAs, numerous snoRNAs are also downregulated in the dnMBC group. SnoRNAs play a role in the post-transcriptional modification and maturation of ribosomal RNAs (rRNAs). They consist of 60–300 nucleotides and are divided into two classes: C/D-box and H/ACA-box snoRNAs [55]. SnoRNAs have not been extensively studied in breast tumors. In the literature, only a few snoRNAs have been highlighted in relation to cancer, which means that extensive research is still needed in order to find out which particular role(s) they play in carcinogenesis. SNORA38B was reported to have a potential role in the PI3K-AKT/ERK/mTOR pathway in breast cancer [55,56]. Luo et al. found that *SNORD3A* is decreased in breast cancer as a result of the downregulation of the transcription factor Meis 1 [57]. In addition, SNORD3A is believed to be a competing endogenous RNA (ceRNA) by acting as a molecular sponge for microRNAs, thus regulating gene expression at the post-transcriptional level [57]. In our study, the snoRNAs *SNORA38B* and *SNORD3A* were significantly downregulated in dnMBC, suggesting they play a role in de novo breast cancer metastasis. Lastly, multiple differentially expressed regulatory genes from our study were identified as pseudogenes, i.e., non-functional copies of protein-encoding genes that have been considered “junk” DNA for many years [58,59]. However, recent studies have highlighted the potential role of expressed pseudogenes in cancer progression [58]. *DUSP5P1* was found to be highly expressed in gastric cancer [60] and was linked to poor prognosis in multiple myeloma [61]. Its protein-encoding counterpart, DUSP5, inhibits the ERK pathway. The *DUSP5P1* pseudogene might interfere with the activity of DUSP5, thus perturbing ERK signaling [62]. *DUSP5P1* was the only pseudogene described in the literature that appeared significant our results. The other pseudogenes (n = 22), all showing very highly significant differential expression between dnMBC and eBC, have not yet been studied in cancer, which underscores the importance and high need to profoundly investigate this type of genes in the cancer setting.

## 5. Limitations of the Study

Our study has some limitations. Firstly, the sample size was limited, which is mostly attributable to the fact that we applied stringent selection and matching criteria. Therefore, further validation of our findings is definitely required. Nevertheless, we think that this stringent selection allows a robust exploratory study. We thought it was important to exclude rare subtypes (often with very specific biology) and only focus on IBC-NST. Matching was adequate, except for tumor size and nodal status, which were a bit more advanced for the dnMBC group. This was inevitable because of an insufficient number of available patients with exactly the same cT/N stage (despite having access to a database of >18,000 patients). Potential imbalances due to imperfect matching were handled through multivariable analyses adjusted for the confounding factors. It should be pointed out that we could only work with patients with sufficient residual tumor tissue from diagnostic core biopsies. To avoid technical bias due to the use of different types of specimens (e.g., biopsy specimens for dnMBC patients versus surgical resection specimens for eBC patients), we employed core needle biopsies for both groups. This seriously hampered the possibility to recruit a substantial number of patients, especially those with de novo disease being limiting. The relatively small cohort size could possibly result in a biased interpretation of the study results, which may not be representative for the entire breast cancer population. Of note, given the sample rarity, we deliberately focused on the most frequent subtype of breast cancer patients, i.e., ER+/HER2− IBC-NST, in order to decrease heterogeneity within the cohort and maximize the chance of detecting significant transcriptomic differences between dnMBC and eBC. Dedicated research will be needed to characterize the other subtypes.

Finally, we only studied the primary tumors of dnMBC patients and not the metastatic lesions, as CNBs from the latter are only rarely available. As a consequence, our results do not reflect the whole tumor microenvironment of de novo metastasized luminal breast cancer.

## 6. Conclusions

This is the first study to highlight the transcriptomic and tumor microenvironmental differences between dnMBC and eBC. Our data suggest that diverse interrelated networks, such as hypoxia-related genes, immunity-related genes, microRNAs, snoRNAs, and pseudogenes are involved in primary breast cancer metastasis. Moreover, after correction for tumor size and grade, the results still showed a similar trend. This study solely explored the biological drivers of primary metastasis in the hormone-sensitive breast cancer subtype at the transcriptomic level. However, if externally validated, these observations may also have clinical implications with regard to diagnostic practice. For patients with tumors exhibiting the ‘non-metastasized’ transcriptomic profile, standard staging procedures at breast cancer diagnosis (e.g., limited to chest radiography, liver ultrasound and/or bone scintigraphy) could still be followed. If, however, a newly diagnosed tumor would exhibit the ‘de novo metastasized’ transcriptomic profile, more exhaustive and sensitive imaging (e.g., PET scan and/or MRI) may be justified, since the accurate detection and localization of distant metastases already existing at tumor diagnosis, is an essential determinant for treatment decisions. The validation of our findings on large external/independent clinical cohorts, as well as further exploration of the disclosed pathways in animal models using breast cancer cell lines with knock-out of specific genes, are necessary to fully understand the biological relevance of the results presented here. Future research should also include the evaluation of the molecular landscape in dnMBC both in the primary tumor and synchronic distant metastases to explore and compare the biology of metastases to matched primary tumors.

## Figures and Tables

**Figure 1 cancers-15-04341-f001:**
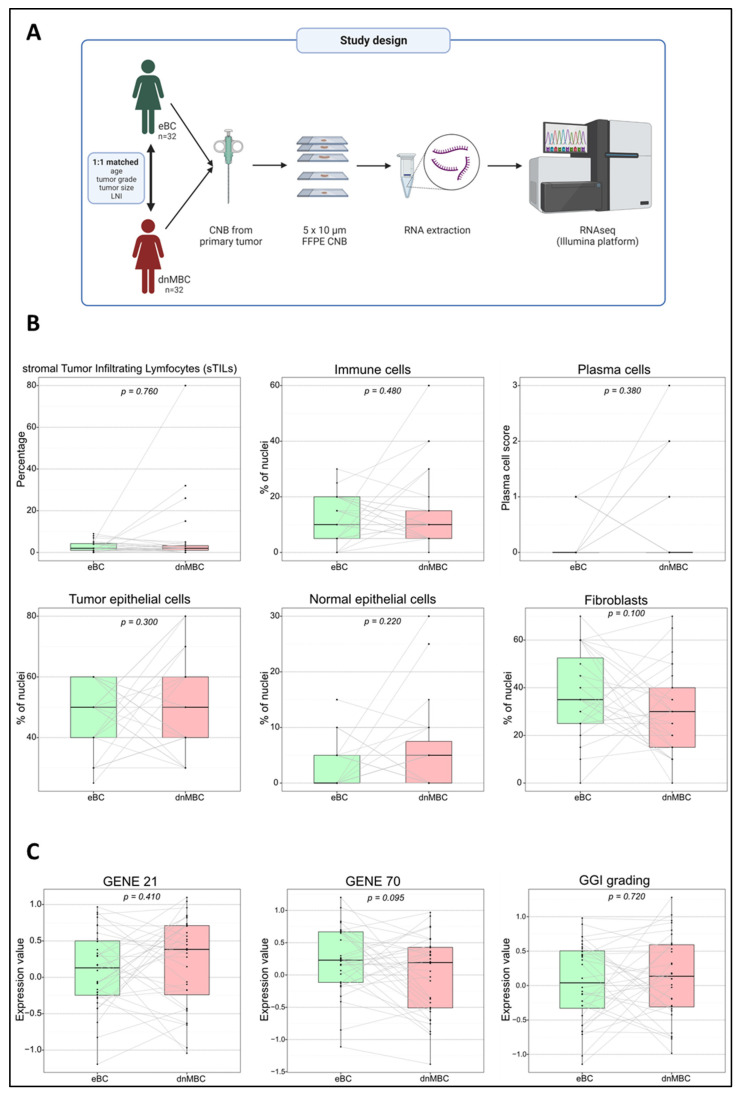
The study design and an overview of the comparable primary tumor CNBs based on pathology and gene expression signatures. (**A**) The study design includes 32 matched patients pairs (dnMBC vs. eBC), where RNA was extracted from the primary tumor CNB and sequenced using the Illumina platform. (**B**) The pathological parameters that were used (sTILs, immune cells, plasma cells, tumor epithelial cells, normal epithelial cells, and fibroblasts) to determine if primary CNBs from both groups were comparable. One extra consecutive FFPE CNB slide was H&E stained and reviewed by an expert breast pathologist to ensure comparable tumor cellular composition across the entire cohort. (**C**) Paired Wilcoxon was used to compare gene expression signatures (GENE21, GENE70, and GGI grading) between the dnMBC and eBC group. CNB: core needle biopsy; dnMBC: de novo metastasized breast tumor group; eBC: non-primary metastatic breast tumor group; FFPE: formalin-fixed paraffin-embedded; GGI: genomic grade index; H&E: hematoxylin and eosin; sTILs: stromal tumor-infiltrating lymphocytes.

**Figure 2 cancers-15-04341-f002:**
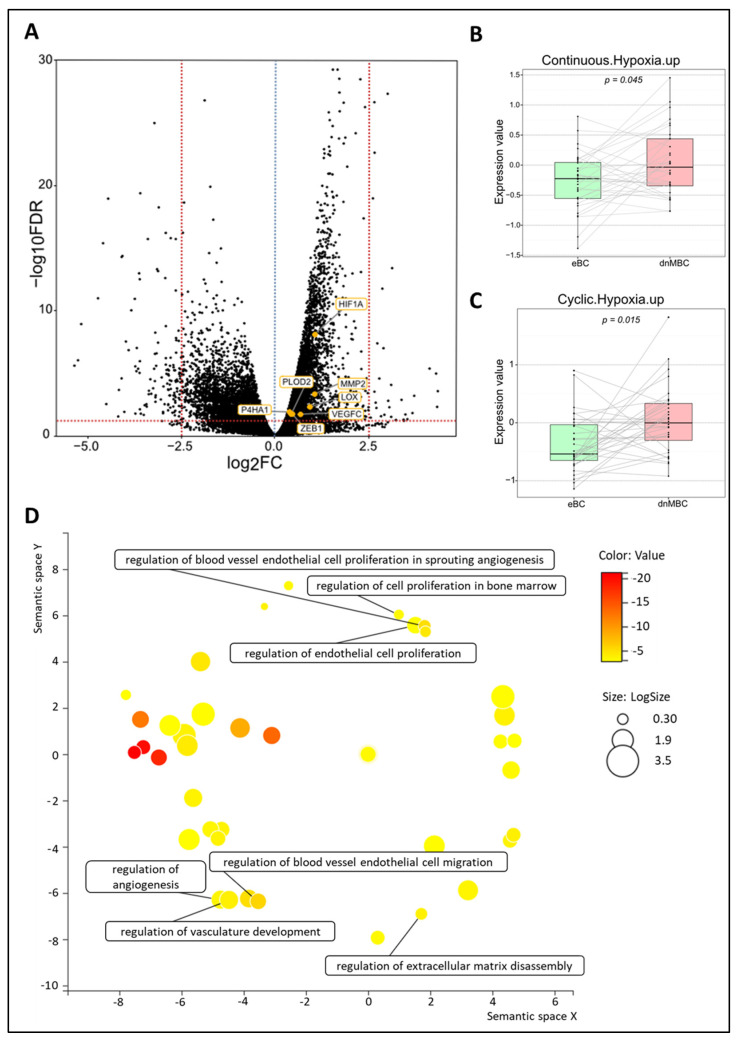
Transcriptomic analysis reveals that hypoxia-related pathways are upregulated in de novo metastasized tumors. (**A**) Volcano plot of differentially expressed genes shows a statistically significant higher expression of seven hypoxia-related genes (*HIF-A*, *PLOD2*, *MMP2*, *LOX*, *VEGFC*, *P4HA1*, and *ZEB1*). A dotted blue line marks a log_2_FC value of zero. A dotted red line crossing the y-axis marks a negative log_10_FDR value of 1.3, which is the transformed FDR-corrected *p*-value of 0.05. A dotted red line on the x-axis marks log_2_FC values of 2.5 and −2.5, respectively. (**B**,**C**) Integrated boxplots of signatures Continuous.Hypoxia.up and Cyclic.Hypoxia.up are upregulated in the dnMBC tumor group compared to eBC tumor group. *p*-values are FDR-corrected. (**D**) GO enrichment analysis visualized in REVIGO displays many mechanisms that are involved in the hypoxia pathway. The terms that are highlighted have a linkage with hypoxia only, because this REVIGO plot represents all the GO terms described in our selected significant DEG dataset. Value stands for the *p*-value alongside the GO term ID from our input dataset. The *p*-values are transformed to Log10 (*p*-value). Size stands for the Log10 (number of annotations for GO Term ID in human species in the EBI GOA database). DEG: differentially expressed gene; dnMBC: de novo metastasized breast tumor group; eBC: non-primary metastatic breast tumor group; FC: fold change; FDR: false discovery rate; GO: gene ontology.

**Figure 3 cancers-15-04341-f003:**
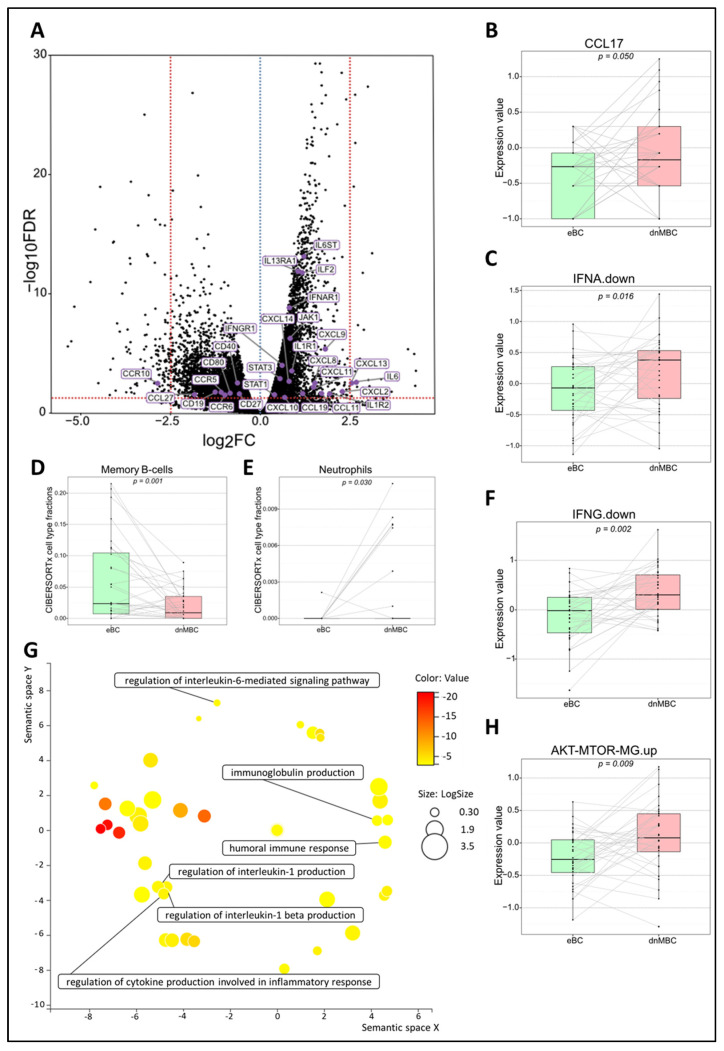
Transcriptomic analysis reveals that immune-related pathways are altered between both study cohorts. (**A**) Volcano plot of differentially expressed genes shows a statistically significant upregulation of multiple chemokines and genes related to the JAK–STAT pathway, while genes of several chemokine receptors and CD-related genes of memory B-cells are significantly downregulated in dnMBC compared to eBC. A dotted blue line marks a log_2_FC value of zero. A dotted red line crossing the y-axis marks a negative log_10_FDR value of 1.3, which is the transformed FDR-corrected *p*-value of 0.05. A dotted red line on the x-axis marks log_2_FC values of 2.5 and −2.5, respectively. (**B**,**C**,**F**,**H**) Integrated boxplots of signatures IFNA.down, IFNG.down, CCL17, and AKT-mTOR-MG.up are significantly upregulated in de novo metastasized tumors. *p*-values are FDR-corrected. (**D**,**E**) Paired Wilcoxon analysis of cell-type fractions in CIBERSORTx software revealed that neutrophils were significantly upregulated, while memory B-cells were significantly downregulated in dnMBC vs. eBC. (**G**) GO enrichment analysis visualized in REVIGO displays many mechanisms that are involved in the immune status of the tumors. The terms that are highlighted have a linkage with immunity only, because this REVIGO plot represents all the GO terms described in our selected significant DEG dataset. Value stands for the *p*-value alongside the GO term ID from our input dataset. The *p*-values are transformed to Log10 (*p*-value). Size stands for the Log10 (number of annotations for GO Term ID in human species in the EBI GOA database). CD: cluster of differentiation; DEG: differentially expressed gene; dnMBC: de novo metastasized breast tumor group; eBC: non-primary metastatic breast tumor group; FC: fold change; FDR: false discovery rate; GO: gene ontology.

**Figure 4 cancers-15-04341-f004:**
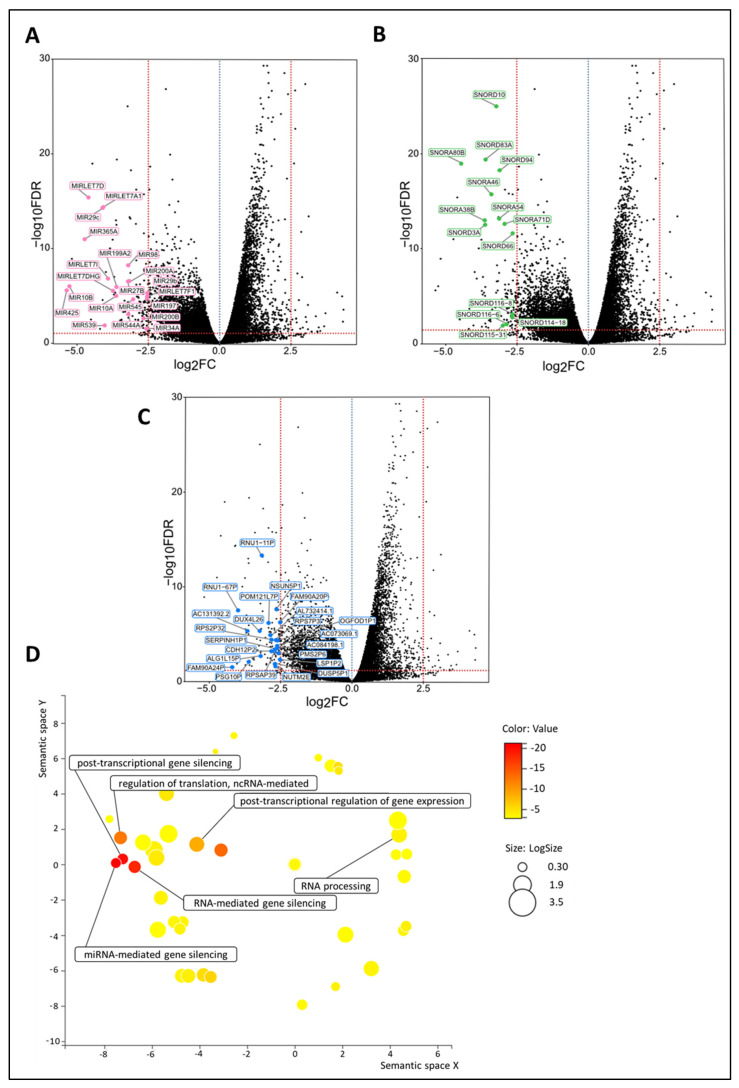
Regulatory genes (i.e., snoRNAs, microRNAs, and pseudogenes) are upregulated in de novo metastasized luminal breast tumors. (**A**–**C**) Volcano plot of differentially expressed genes shows a statistically significant upregulation of multiple snoRNAs (n = 14), microRNAs (n = 21), and pseudogenes (n = 23) in tumors of dnMBC compared to tumors from eBC. A dotted blue line marks a log_2_FC value of zero. A dotted red line crossing the y-axis marks a negative log_10_FDR value of 1.3, which is the transformed FDR-corrected *p*-value of 0.05. A dotted red line on the x-axis marks log_2_FC values of 2.5 and −2.5, respectively. (**D**) GO enrichment analysis visualized in REVIGO displays many gene regulatory mechanisms that are involved in the de novo tumors. The terms that are highlighted have a linkage with regulatory genes only, because this REVIGO plot represents all the GO terms described in our selected significant DEG dataset. Value stands for the *p*-value alongside the GO term ID from our input dataset. The *p*-values are transformed to Log10 (*p*-value). Size stands for the Log10 (number of annotations for GO Term ID in human species in the EBI GOA database). DEG: differentially expressed gene; dnMBC: de novo metastasized breast tumor group; eBC: non-primary metastatic breast tumor group; FC: fold change; FDR: false discovery rate; GO: gene ontology; snoRNA: small nucleolar RNA.

**Figure 5 cancers-15-04341-f005:**
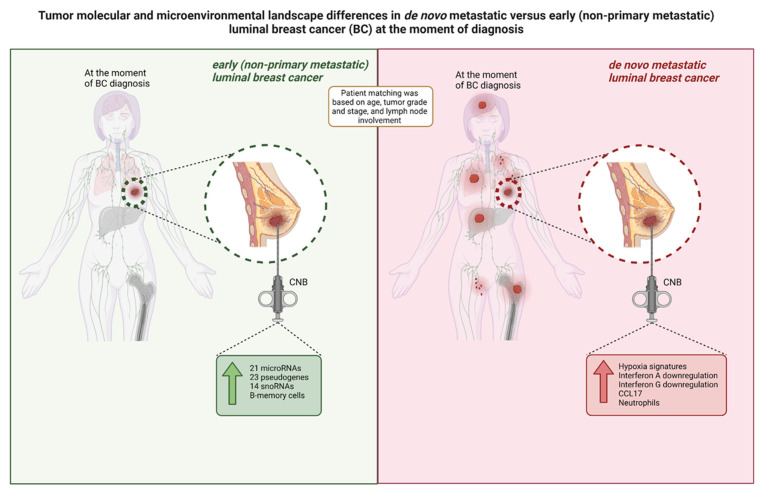
Overview of the tumor molecular and microenvironmental landscape differences in primary CNB tumors from luminal breast cancer patients within our study cohort. De novo metastatic luminal breast cancer tumors exhibit a higher expression of hypoxia signatures, immunity-related signatures (Interferon A downregulation, Interferon G downregulation, and CCL17) and neutrophils at diagnosis, while tumors from patients with non-primary metastatic luminal BC exhibit a higher expression of regulatory genes (i.e., microRNAs, pseudogenes, and snoRNAs) and memory B-cells. BC: breast cancer; CNB: core needle biopsy; snoRNA: small nucleolar RNA.

**Table 1 cancers-15-04341-t001:** Patient characteristics (age at diagnosis) and tumor properties (tumor grade and size, receptor status, lymph node involvement, and location of relapse). The matching criteria were based on age, tumor size and grade, and lymph node involvement. Exploratory *p*-values were inserted to indicate matching effectiveness between the two groups. *p*-values were assessed with Fisher’s exact test and paired Wilcoxon tests for categorical and continuous variables, respectively.

Variables	Statistics	De Novo Metastasized BC Group (dnMBC)	Non-Primary Metastasized BC Group (eBC)	*p*-Values
Age patients				0.532
	N	32	32	
	Median	62	61	
	Average	61.69	60.84	
	Range	[32.0; 88.0]	[36.0; 83.0]	
Grade of tumor				1.000
Grade 2	n/N (%)	14/32 (44%)	15/32 (47%)	
Grade 3	n/N (%)	18/32 (56%)	17/32 (53%)	
Receptor status				0.672
ER+/HER2−/PR+	n/N (%)	28/32 (87%)	30/32 (94%)	
ER+/HER2−/PR−	n/N (%)	4/32 (13%)	2/32 (6%)	
Clinical staging (cT)				*0.009*
cT1	n/N (%)	1/32 (3%)	6/32 (19%)	
cT2	n/N (%)	17/32 (53%)	23/32 (72%)	
cT3	n/N (%)	4/32 (13%)	3/32 (9%)	
cT4	n/N (%)	10/32 (31%)	0/32 (0%)	
cT4b	n/N (%)	3/32 (9%)	0/32 (0%)	
cT4c	n/N (%)	1/32 (3%)	0/32 (0%)	
cT4d	n/N (%)	5/32 (16%)	0/32 (0%)	
Lymph node involvement (cN)				<*0.001*
cN0	n/N (%)	6/32 (19%)	21/32 (66%)	
cN1	n/N (%)	11/32 (34%)	11/32 (34%)	
cN2	n/N (%)	3/32 (9%)	0/32 (0%)	
cN3	n/N (%)	12/32 (38%)	0/32 (0%)	
Tumor size (mm)				<*0.001*
	Median	37	27	
	Average	43.68	28.47	
	Range	[16.0; 140.0]	[15.0; 55.0]	
Location of metastasis				-
Brain	n/N (%)	0/32 (0%)	-	
AbdominalNonLiver	n/N (%)	3/32 (9%)	-	
Liver	n/N (%)	13/32 (41%)	-	
Cutaneous	n/N (%)	3/32 (9%)	-	
Lung	n/N (%)	11/32 (34%)	-	
Bone	n/N (%)	21/32 (66%)	-	
Locoregional lymph nodes	n/N (%)	12/32 (38%)	-	
Others	n/N (%)	1/32 (3%)	-	

*p*-values less than 0.05 are reported in italics.

## Data Availability

The dataset used and analyzed during the current study is available from the corresponding author on reasonable request.

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
