# Peer review of "Differences in the Tumor Molecular and Microenvironmental Landscape between Early (Non-Metastatic) and De Novo Metastatic Primary Luminal Breast Tumors"

_cancers, 2023, doi:10.3390/cancers15174341_

Round 1

Reviewer 1 Report

The authors present a comparative biological analysis of a small series of breast cancers, aiming at deciphering the differences between cancers with de novo metastases (dnBC) and those without metastases at diagnosis (eBC)

Although interesting, this report will be improved by some important revisions

MAJOR

1. the authors claim this is a matched-pair analysis, where dnBCs are matched with eBCs sharing the same stage and core biological features (T, N, ER, Grade etc). As seen in the population table, this is not true, with important differences regarding tumor size, PR (or ER ? not clear in the table !) expression, grade. Exploratory p values should be indicated in the table.

2. consequently, the whole analysis might be hampered by this limitation. It is not sure that Wilcoxon testing may be the more appropriate statistical approach. Mann Whitney testing may be adequate. This very important issue should addressed in the discussion section

3. A principal component analysis based on the RNAseq data should be presented, in order the explore the global difference between the two subsets (dnBC and eBC)

4. Pathologic assessment was performed with eyeballing, which appears a very simplistic approach. Was it performed centrally ? by how many observers ? We strongly suggest to complement by immunohistochemical studies. The authors should also be aware that cellular deconvolution may be performed through RNAseq analysis. PLease expand these data.

MINOR

5. Again in the population table, it is not clear what "lymph nodes" refer to: locoregional lymph nodes ? distant lymph nodes ? please specify

6. Typo : italics lines 218 sqq

7. No data on the molecular landscape of metastases are shown. We understand this was not the purpose of the study, however it would greatly enhance the results to explore and compare the biology of metastases to matched primary tumors. The authors should discuss this issue as well.

Reviewer 2 Report

The article by Yentl Lambrechts et al. entitled " Differences in the tumor molecular and microenvironmental land-scape between early (non-metastatic) and de novo metastatic primary luminal breast tumors" is quite interesting. However, it still raises the following issue.

1. The background is too general and lacks specific details about the research gap and significance of the study.

 2. The sample size is small and may not be representative of the entire population.

3. The methods lack details about the selection criteria for the eBC group and the matching process.

4. Authors should include statistical analysis in 'Materials and methods'. The statistical analysis is limited to a paired Wilcoxon test and does not account for potential confounding factors.

5. The results are difficult to interpret due to the lack of functional analysis and validation of the differentially expressed genes.

6. Authors should state the limitations of the study in a separate title

7. The conclusion is not supported by the results and does not provide any implications for clinical practice or future research.

Minor editing of English language required

Round 2

Reviewer 1 Report

the revised form is acceptable

Reviewer 2 Report

After addressing the reviewer's constructive feedback, the authors have made necessary revisions to the paper, rendering it ready for publication. But, it should be noted that the manuscript has not been formatted according to the journal's guidelines and template.

Minor editing of English language required